# The Medial Prefrontal Cortex as a Central Hub for Mental Comorbidities Associated with Chronic Pain

**DOI:** 10.3390/ijms21103440

**Published:** 2020-05-13

**Authors:** Kai K. Kummer, Miodrag Mitrić, Theodora Kalpachidou, Michaela Kress

**Affiliations:** Institute of Physiology, Medical University of Innsbruck, 6020 Innsbruck, Austria; kai.k.kummer@gmail.com (K.K.K.); m.mitric@vu.nl (M.M.); theodora.kalpachidou@i-med.ac.at (T.K.)

**Keywords:** infralimbic cortex, prelimbic cortex, cholinergic synapse, anterior cingulate cortex, neuropathic pain, mental comorbidities, depression, GABAergic signaling

## Abstract

Chronic pain patients frequently develop and suffer from mental comorbidities such as depressive mood, impaired cognition, and other significant constraints of daily life, which can only insufficiently be overcome by medication. The emotional and cognitive components of pain are processed by the medial prefrontal cortex, which comprises the anterior cingulate cortex, the prelimbic, and the infralimbic cortex. All three subregions are significantly affected by chronic pain: magnetic resonance imaging has revealed gray matter loss in all these areas in chronic pain conditions. While the anterior cingulate cortex appears hyperactive, prelimbic, and infralimbic regions show reduced activity. The medial prefrontal cortex receives ascending, nociceptive input, but also exerts important top-down control of pain sensation: its projections are the main cortical input of the periaqueductal gray, which is part of the descending inhibitory pain control system at the spinal level. A multitude of neurotransmitter systems contributes to the fine-tuning of the local circuitry, of which cholinergic and GABAergic signaling are particularly emerging as relevant components of affective pain processing within the prefrontal cortex. Accordingly, factors such as distraction, positive mood, and anticipation of pain relief such as placebo can ameliorate pain by affecting mPFC function, making this cortical area a promising target region for medical as well as psychosocial interventions for pain therapy.

## 1. Introduction

Functional magnetic resonance imaging (fMRI) studies in both humans and rodents have shown that there is not a single brain structure responsible for pain perception, but that different cortical, subcortical, and associative brain regions are activated during a painful stimulus [1]. Due to its complexity and interconnectedness, some researchers refer to this network of regions as the “pain matrix” [2]. Most commonly activated brain regions in response to pain are the thalamus, the somatosensory cortices, the insular and the prefrontal including the anterior cingulate cortex (ACC), the lentiform nucleus, cerebellum, amygdala, and the nucleus accumbens [1,3,4,5,6]. The thalamus relays nociceptive information to the insula, ACC and somatosensory cortices to process sensory-discriminative properties, including the location, quality, and intensity of a pain stimulus, while other brain regions such as the prefrontal cortex, insular cortex, limbic system, cerebellum or periaqueductal gray (PAG) play a role in affective components of the pain sensation and the formation of pain memory [7,8,9,10,11,12,13,14]. 

Its predominant role in aversive learning and memory puts the medial prefrontal cortex (mPFC) in a pole position as a hub for the development of mental comorbidities associated with chronic pain. Neurons within the mPFC respond to noxious stimuli [15] and electrical stimulation of mPFC inhibits nociceptive responses [16]. Human brain fMRI studies show activation of mPFC during the perception of acute pain stimuli and functional deactivation in chronic pain patients (for review see [17]). Affective and cognitive components of pain sensations are processed by the mPFC [6,18]. Furthermore, the basis of placebo induced analgesia or expectancy of pain relief is based on the ACC-PFC-PAG descending pain pathway, which influences endogenous opioid activity (reviewed by [6]). Resting-state network studies in chronic pain patients report disrupted network properties within the interacting brain regions, the default mode network (DMN). This also includes reduced connectivity between the mPFC and the posterior regions of the DMN, as well as increased connectivity between the mPFC and the insular cortex, which is correlated to pain intensity [19]. In rodents, the mPFC is primarily known for its prominent role in attention, working memory, and goal-directed behavior [20]. It provides top-down regulation of sensory and affective processes, including inhibition of both sensory and affective pain signals by descending projections to the various brain and spinal cord regions: its projections are the main cortical input of the PAG, which is part of the descending inhibitory pain control system at the spinal level [21,22,23]. In both human subjects and rodent models, the mPFC undergoes structural as well as functional changes in chronic pain states [24,25,26,27,28,29,30,31], which is reflected by cognitive deficits and decreased attention (for review see [32]). This review aims to dissect the main local circuitry, including transmitter systems, together with input and output connectivity of mPFC subregions and their importance for affective pain processing and the mental comorbidities frequently associated with chronic pain disorders.

## 2. Input Systems to mPFC

Neuroanatomical tracing studies reveal distinct mPFC subregions, however, nomenclature and conventions on how to apply them differ between stereotaxic atlases [33,34,35]. Here, we address the prelimbic cortex (PrL), the infralimbic cortex (IL), and the anterior cingulate cortex (ACC) as mPFC subregions, which correspond to the human pregenual and subgenual ACC (Brodmann Areas 32 and 25), and the midcingulate cortex (Brodmann Area 24), respectively [35]. Incoming signals are received by the ACC indirectly via the medial thalamus as well as the amygdala, primary somatosensory cortex, and insular cortices [36,37,38,39,40,41]. Projections from the agranular insular cortex, basal forebrain nuclei, basal nuclei of the amygdala, midline thalamic nuclei (including the paraventricular nucleus), the hippocampus CA1 region and subiculum, ventral tegmental area, PAG, dorsal raphe nuclei, and locus coeruleus target both PrL and IL [42]. Virtually no input to the two regions is documented from primary motor and somatosensory cortices, as well as the dorsal striatum and nucleus accumbens. However, while the PrL receives inputs mainly from anterior forebrain regions, the IL is preferentially targeted by projections from anterior hypothalamus, medial and lateral septum, substantia innominata, as well as the dorsolateral tegmentum [42]. Cholinergic input to the mPFC arises from the basal forebrain (BF) and is associated with attentional tasks and cognitive performance [20]. The vertical limb (VDB) and the hind limb (HDB) of the diagonal band of Broca (DB) are generally accepted as cholinergic input regions to the mPFC [43,44,45,46,47]. Recent studies indicate cholinergic projections from the medial septum (MS) to the mPFC which might contribute to the cognitive and emotional impairments induced by pain [48,49,50]. MS inactivation concurrently reduces the activation of the prefrontal cortex and inhibition of cholinergic MS neurons projecting to the ACC alleviates pain-induced anxiety [50,51,52]. 

## 3. Neuron Types and Circuitry in mPFC Cortical Layers

Within the mPFC, five cellular layers can be distinguished [53]: excitatory pyramidal neurons in layers 2/3 (L2/3) and L5 make up ~80% of neurons and (mainly inhibitory) interneurons the remaining ~20% [54]. The mPFC receives inputs and sends reciprocal projections back to almost all areas of the sensory cortex, various motor structures, and subcortical regions, including the hippocampus and amygdala [55]. The majority of inputs are received in superficial L2/3, whereas outputs from layers L2/3 are shared between the prefrontal subregions and also terminate in deeper layers L5 and L6, from where efferent projections are sent to various subcortical regions [54,56]. 

### 3.1. Pyramidal Neurons 

Pyramidal neurons differ between superficial and deep layers regarding morphological features, molecular markers as well as responsiveness to sensory stimuli; they receive unique synaptic inputs but also possess unique action potential (AP) firing patterns and physiological phenotypes [57,58,59,60,61]. In superficial layers L1 and L2/3, pyramidal neurons and interneurons can reciprocally connect in every possible way [62]. Different layers are tightly coupled to specific functions and differently affected in mental disorders, including schizophrenia, bipolar disorder, stress-induced depression, and substance abuse [63,64,65,66,67]. Pyramidal neurons in L2/3 are nociceptive and show regular spiking, intermediate firing, or intrinsic bursting [68,69]. Excitatory pyramidal neurons in layers L2/3 and L5 respond to noxious mechanical and thermal stimuli with an increase in firing rates and are related to the unpleasantness induced by acute pain [69,70,71,72,73,74]. Pain conditions including peripheral nerve injury induce long-lasting synaptic plasticity by different variations of long-term potentiation (LTP) or long-term depression (LTD) to increase excitatory synaptic transmission resulting in hyperexcitability and disinhibition of the ACC, whereas inhibitory transmission is reduced [68,75,76,77,78,79,80]. 

### 3.2. Inhibitory Interneurons

Inhibitory GABAergic interneurons make up 10–20% of all mPFC neurons and are found in all layers [81]. GABAergic interneurons are targeted by thalamocortical projections [82], and as they are divided based on neuropeptide markers together with their layer-specific axonal projections (L1, L2/3, L5, and white matter-projecting interneurons [83]), they may play differential roles in maintaining the excitation/inhibition balance in different mPFC layers and underlie previously described laminar differences in pain processing [84].

#### 3.2.1. Parvalbumin Interneurons

Fast-spiking parvalbumin (FS-PV+) interneurons uniformly show increased and sustained firing during goal-driven attentional processing, correlating to the level of attention. Successful allocation of attention is characterized by strong synchronization of FS-PV+ neurons, increased gamma oscillations, and phase locking of pyramidal firing. FS-PV+ neurons thus act as a functional unit coordinating the activity in the local mPFC circuit during goal-driven attentional processing [85]. PV+ interneurons are critically important for fear conditioning and the reward system, and successful memory consolidation also requires coherent hippocampal-neocortical communication mediated by PV+ cells [86,87]. Optogenetic activation of PV+ interneurons in neuropathic mice causes an increase in mechanical hypersensitivity, suggesting that these cells may be directly involved in sensory processing of chronic pain [88]. Bilateral morphological changes in PV+ interneurons and layer L5/6 IL pyramidal neurons are correlated with neuropathic pain-related cognitive impairments [89] (Figure 1).

#### 3.2.2. CCK Interneurons

Cholecystokinin (CCK) interneurons comprise a larger proportion of the mPFC interneurons compared to PV+ interneurons, targeting a wide range of neuronal subtypes with a distinct connectivity pattern and CCK+, but not PV+ interneurons play a critical role in the retrieval of working memory [91]. Although their functional importance is less understood, they are associated with emotional and cognitive symptoms of major depressive disorder [92].

#### 3.2.3. Somatostatin Interneurons

Somatostatin interneurons facilitate hippocampal-prefrontal synchrony and prefrontal spatial encoding and contribute to cognitive performance. In the prefrontal cortex, they control affective state discrimination [93,94].

#### 3.2.4. VIP Interneurons

GABAergic vasoactive intestinal peptide (VIP) neurons inhibit PV+ and somatostatin-expressing interneurons, thereby disinhibiting pyramidal neurons [95,96,97].

### 3.3. Excitatory Interneurons 

Neocortical choline acetyltransferase (ChAT)-expressing interneurons are a subclass of VIP+ interneurons of which circuit and behavioral functions are largely unknown. ChAT-VIP+ neurons directly excite neighboring neurons in several layers through the fast synaptic transmission of acetylcholine (ACh) in mPFC and control attention behavior [98].

### 3.4. Modulation by Glial Cells

The modulation of electrical and synaptic neuronal activity by glial cells in pain states is well accepted, whereas the role of microglia and astrocytes in the mPFC is only starting to gain interest [99,100,101]. The majority but not all studies support an upregulation of microglia or microglial markers, that might at least in part be induced by interleukin 1 beta (IL-1β) [102,103,104,105,106,107,108,109,110,111,112]. Inhibition of microglial activation in neuropathic pain models alleviates mechanical allodynia in nerve ligated mice and affective components of pain-like behavior [103,107]. Interestingly, under healthy conditions, the nociceptor transducer channel TRPV1 is specifically expressed in microglia in the ACC, where it controls microglial activation and indirectly enhances glutamatergic transmission in neurons [113]. In addition, astrocyte marker expression increases in inflammatory and neuropathic pain models at least partially as a consequence of enhanced glutamate levels in the ACC [107,114,115,116,117,118,119,120]. Astrocyte numbers appear to be relatively constant, however, complex changes in glutamate transporter GLT1 and ionotropic as well as metabotropic glutamate receptor subunits take place in astrocytes upon peripheral painful events such as formalin injection or paclitaxel neuropathy [120,121]. The selective ablation of astrocytes inhibits long-term facilitation associated with inflammatory pain and relieves escape avoidance behavior, which stresses their functional importance for pain processing in the ACC [117,118].

## 4. Local Circuits and Differences Between Cortical Layers 

### 4.1. Layer L1

Like in other neocortical areas, layer L1 contains only a few neurons, which are almost exclusively GABAergic interneurons, predominantly late-spiking, continuously adapting and inhibiting pyramidal neurons [57,122,123,124,125]. L1 interneurons are a major target of neuromodulatory inputs that can influence circuit computations by shifting the balance between excitation and inhibition [126]. They express a mixed population of nicotinic receptors, and are densely innervated by both extracortical and recently discovered intracortical cholinergic inputs [98,124]. Even though L1 is largely devoid of neuronal cell bodies, it is rich in spiny apical arborizations of dendrites originating in deeper cortical layers. As such, L1 has been identified as a prominent feedback pathway receiving glutamatergic projections from higher cortical areas as well as various thalamic nuclei [125,127]. Thalamic inputs in the PFC preferentially target L1 neurons and trigger feed-forward inhibition in neighboring interneurons as well as L2/3 pyramidal cells [128,129]. They may, therefore, act as a potential gate of prefrontal pain processing. 

In the primary somatosensory cortex (S1) of neuropathic mice, dendritic apical branches of pyramidal neurons show increased spine motility and apical tufts exhibit increased dendritic Ca^2+^ spikes within days after surgery and this is preserved for almost a month [130,131]. Similar processes apply to the mPFC, and chronic stress exposure induces short- and long-term synaptic changes, in particular atrophy and reduction in spine density of L5 pyramidal cell dendrites [132,133]. Likewise, apical dendrites of both L2/3 and L5 mPFC pyramidal neurons undergo alterations in dendritic morphology and length after nerve injury [84]. Chronic constriction injury reduces expression of the hyperpolarization-activated and cyclic nucleotide-gated HCN1 channel in layer L1 of the ACC [134].

### 4.2. Layer 2/3

Pyramidal neurons of layer L2 show a unique morphological phenotype with a wide span of apical and a particularly narrow span of basal dendrites [135]. Dendritic spines of L2/3 pyramidal neurons receive mostly excitatory glutamatergic projections from the midline thalamic nucleus, ventral hippocampus, basolateral amygdala (BLA), as well as the contralateral mPFC, but only L3 pyramidal neurons generate action potential bursts in response to depolarizing currents [58,135,136]. L2/3 pyramidal neurons belong to the group of intratelencephalic neurons, targeting the ipsilateral perirhinal cortex, amygdala, and striatum as well as the contralateral striatum and cortex [137,138]. They also influence the local circuitry by providing monosynaptic excitation of L5 pyramidal neurons, as well as by targeting GABAergic interneurons, highlighting their importance in balancing excitatory and inhibitory transmission [139]. Increased NMDA/AMPA ratio, as well as higher complexity and spine density of basal dendrites in ACC/PrL L2/3 pyramidal neurons, is observed one week after surgery and can be linked to sprouting of noradrenergic fibers in L2/3 and subsequent alpha2 induced modulation of I_h_ currents [25,140]. Increased input resistance and depolarized membrane potential, together with enhanced glutamatergic release are reported in PrL L2/3, whereas IL shows increased length and complexity of only L2/3 but not L5 pyramidal neuron dendrites, suggesting regional and laminar dichotomy [75,84,141,142,143,144]. The amount of excitatory/inhibitory synaptic connections is bidirectionally diminished, resulting in a misbalanced excitatory/inhibitory ratio, and apoptotic loss of subpopulations of inhibitory neurons has been described [79,145]. 

The ACC is the most dorsal subregion of the mPFC and often studied separately due to its prominent role in the affective component of pain processing [146]. Neuropathy enhances presynaptic glutamate release, postsynaptic AMPA mediated response as well as excitability of ACC layer L2/3 pyramidal neurons, in particular those which are intermediately firing [68,73,75,144]. Apart from augmented glutamatergic transmission, prolonged nociceptive sensitization during the initial induction phase can weaken inhibitory control in layer L2/3 as indicated by reduced frequency of miniature and spontaneous inhibitory postsynaptic currents, suggesting presynaptic GABAergic plasticity [147]. In addition, long-term depression (LTD) can be impaired [148,149]. Lastly, ex vivo findings are supported by in vivo electrophysiological measurements, as long-term changes in spontaneous membrane oscillations are observed in the ACC layer L2/3 after nerve injury [150].

### 4.3. Layer 5

L5 pyramidal neurons differ from L2/3 pyramidal neurons by a thick band of thalamocortical fibers present in deep L3, as well as by their larger cell bodies [135,138,151]. In contrast to sensory cortices, thalamic afferents in deeper cortical layers are sparse. However, long apical dendrites of L5 pyramidal neurons climb towards layer L1 and form highly branched apical tufts allowing them to be accessed by thalamic inputs [151]. L5 pyramidal neurons are diverse in terms of their morphological and electrophysiological features [135]. In contrast to layer L2/3, which has long been regarded as the input layer of the PFC, layer L5 represents a typical output layer of the cortex with two projection-specific subpopulations: intratelencephalic neurons, also present and already described in layer L2/3, and pyramidal tract neurons, which project mainly to the ipsilateral striatum, thalamus, or brainstem [126,152]. The two neuron subpopulations differ in their intrinsic electrophysiological properties, ion channel composition, and responsiveness to acetylcholine, noradrenaline, and adenosine [126,152]. 

Evidence for input-specificity of synaptic alterations onto L5 pyramidal and interneurons is emerging from rodent models of sustained pain [153,154,155,156]. ACC L5 pyramidal neurons increase their excitability following nerve injury by modifications of pacemaker ion channels of the HCN family through glutamatergic signaling via metabotropic mGluR1 [74,79,134,154,157]. Even though L5 pyramidal neurons projecting to the contralateral ACC show barely any I_h_ current and express remarkably low levels of HCN1, they still display hyperactivity after nerve injury [152,154,156]. The downregulation of voltage-gated potassium channels of the K_v_2 family, which can be rescued by mGlur1 antagonists, further contributes to this effect [154]. Layer L5 pyramidal neurons are tightly regulated by local inhibitory networks, and an excitatory/inhibitory imbalance may account for the observed changes. Even though the excitability of ACC L5 fast-spiking (FS) interneurons is unaltered, the loss of local synaptic connectivity between pyramidal neurons and these interneurons, together with an imbalance in excitation-inhibition favor excitation in neuropathic pain [79,156,158]. 

In contrast to superficial ACC/PrL and deep IL, pyramidal neurons of PrL layer L5 develop reduced excitability in chronic pain models [68,73,79,84,88,159,160,161]. This is at least in part mediated by enhanced feed-forward inhibition of L5 pyramidal neurons by PV+ interneurons across different layers [88,160]. A long-range BLA to mPFC to PAG to spinal cord pathway links nerve injury to the enforcement of the BLA-driven input onto PrL L5 PV+ interneurons, followed by reduction of PrL L5 pyramidal neuron output to ventrolateral PAG (vlPAG), which compromises the descending modulatory pain control towards the spinal cord [161]. Such changes can also be driven by reduced excitatory input and can be linked to inputs originating in both the mediodorsal thalamus and the ventral hippocampus [153,162]. They stress the differential roles of neuron populations in different layers and mPFC subregions in the processing of chronic pain.

### 4.4. Layer 6

In acute brain slices, L6 is easy to distinguish from L5 since its pyramidal neurons possess smaller cell bodies [135,137,163]. More than a third of L6 pyramidal neurons extend large apical dendrites into layer L1 where they receive extracortical input. In contrast to L5, the dendrites of the L6 pyramidal neurons are not oriented perpendicularly towards the brain surface, but rather are oblique or inverted [135]. L6 receives dense cholinergic innervation, which drives a surprisingly strong nicotinic receptor-dependent excitation of pyramidal neurons [164,165]. This distinguishes them from other pyramidal neurons within the mPFC, making them particularly susceptible to the influence of cholinergic projections or local interneurons (reviewed by [18]). A potential role of layer L6 in chronic pain is supported by the activity-dependent increase in p-ERK 1/2 expression not only in L2/3 and L5, but also in layer L6 of the mPFC two weeks following induction of trigeminal neuropathy [166]. The majority of L6 pyramidal neurons give rise to corticothalamic projections and form a corticothalamic feedback pathway that may influence the reliability of thalamic responses to nociceptive stimuli [42]. Other prominent projections target the lateral hypothalamus, the dorsal striatum, thalamic nuclei, but also cortical microcircuits [16,167,168,169].

## 5. Specific Roles of mPFC Subregions 

A multitude of preclinical studies address the role of the mPFC in general for acute, inflammatory, or neuropathic pain. For example, activation of pyramidal neurons in the mPFC attenuates visceral pain [170]. Inflammatory or neuropathic pain leads to impaired performance in the novel object recognition task, cognitive impairment, anxiety, and depression-like behavior as well as reduced sociability [166,171,172,173]. These functional deficits are associated with the deregulated activity or expression of neurotransmitters, their receptors, or regulatory enzymes within the mPFC [30,104,174,175,176,177,178,179,180].

### 5.1. The Anterior Cingulate Cortex (ACC) 

The anterior cingulate cortex is critically important for the affective component of the pain sensation and pain-related mood disorders [181,182,183,184]. Lesions of the ACC alleviate the affective aspects of pain, decrease escape/avoidance behaviors, and decrease the floating time in the forced swim test in neuropathic pain models [185,186,187,188]. In particular, the rostral ACC appears to be important for processing aversive stimuli and posterior lesions abolish the anti-aversive effect of rostral ACC lesion [189]. Although a dual role of ACC in pain sensitization and affect has been corroborated by differential effects observed with ACC microinjections vs. systemic administration of drugs such as morphine, gabapentin, or T-type calcium channel inhibitors, the mechanistic details regarding its modulatory effect on mechanical and thermal sensitivity, spontaneous pain behavior, inflammatory pain, and formalin-induced mechanical hypersensitivity are not yet fully consistent [190,191,192,193]. Different neuronal subpopulations may be relevant in this context. Optogenetic strategies targeting individual neuronal subpopulations help to dissect the local circuits within the ACC [184,189,194,195,196]. Optogenetic stimulation of Thy-1 expressing ACC neurons enhances anxiety and depressive-like behaviors in mice and in turn, bilateral optogenetic inhibition of ACC results in anti-aversive and anti-anxiodepressive effects [188,197]. Optogenetic silencing of excitatory CaMKII positive neurons induces enhanced exploration in the least preferred compartment in a CPP paradigm, and this is not observed by activation of the inhibitory PV+ neurons [198]. Besides their local mPFC connectivity, ACC neurons synapse onto neurons in the neighboring midcingulate and retrosplenial cortices [199]. Pyramidal neurons located in deeper layers project to the hypothalamus, amygdala, PAG, the contralateral hemisphere or other supraspinal areas as well as directly to the spinal dorsal horn [134,156,183]. Direct excitatory projections to neurons in laminae I-III of the spinal dorsal horn are activated in models of neuropathic and cancer pain [200,201,202,203]. Electrical or chemical activation of the ACC facilitates the tail-flick reflex, supporting the existence of descending facilitation [204]. 

Although there is a general consent that ACC exhibits hyperexcitability in neuropathic pain conditions, there are studies indicating that this generalization may not be absolute. Projections from the mediodorsal thalamus to layer L5 ACC pyramidal neurons elicit both excitation and inhibition, the ratio of which would favor inhibition in a subpopulation of pyramidal neurons that project to subcortical areas [156]. The existence of two different subpopulations of pyramidal neurons in the ACC L5 has been previously reported and the inhibition of the neurons that project to the contralateral side to a nerve injury promotes analgesia without influencing aversive behavior [134]. Inhibiting the excitatory ACC neuronal subpopulation results in aversive behavior during neuropathic pain [156]. 

### 5.2. The Prelimbic (PrL) and Infralimbic (IL) Cortices

Only recently, the specificity of mPFC subregions is receiving more attention [170,205,206,207,208,209]. In particular, electrophysiological recordings of L5 pyramidal cells reveal relevant functional differences between PrL and IL: the GABA_A_ antagonist bicuculline modulates neuron activity in PrL, whereas cannabinoid receptor CB1 and mGLuR5 are relevant components in IL [22,27,210].

Prelimbic cortex neurons show increased firing rates after noxious stimulation, while maintained nociceptive input suppresses both basal spontaneous and pain-evoked activity, which can be ascribed to increased gamma-aminobutyric acid (GABA) and decreased glutamate in PrL/IL [211,212,213,214,215,216]. Neuropathic pain causes disruption of working memory and PrL neuronal firing activity, which is reversed by selective inhibition of PrL pyramidal cells using optogenetic interventions [217]. Already one week after induction of neuropathic pain, L5 pyramidal cells of PrL/ACC show reduced responses to excitatory glutamatergic inputs, without changes in intrinsic excitability [153]. This functional deactivation of PrL pyramidal cells may be attributed to a feed-forward inhibition mediated by PV+ GABAergic interneurons, and modulation of these neurons [88]. Anxiety-like behaviors develop at later stages [218]. In neuropathic pain models, metabotropic glutamate receptors mGluR5 and mGluR7 are upregulated in PrL, and pharmacological interventions suggest opposing roles of these two receptors: while mGluR5 aggravates comorbid pain and negative symptoms, mGluR7 improves cognitive performance and restores excitation/inhibition balance at PrL neurons [219,220]. Optogenetic inhibition of the PrL, or one of its projection targets, the nucleus accumbens, augments sensory and affective symptoms of acute pain, and also increases nociceptive sensitivity and aversive responses in neuropathic pain models [221]. Altogether, deactivation of PrL appears to be associated with affective deficits associated with persistent pain and inhibition of hyperactivity of amygdala neurons reverses the deactivation of PrL and concomitant cognitive deficits [26]. This suggests a general shut-down of PrL as an important pathogenetic mechanism that can be overcome by low-frequency electrical stimulation of the PrL cortex or optogenetic activation of PrL layer L5 neurons to relieve both sensory and affective signatures of acute and sustained pain [211,212,222]. 

In the infralimbic cortex, both mGluR1 and mGluR5 activation induces pronociceptive behavior in monoarthritic rats, which is based on increased activation of dorsal reticular nucleus neurons [223,224]. Peripheral inflammation decreases BDNF levels in the IL but not PrL, and infusion of BDNF into the IL alleviates inflammatory pain and accelerates long-term recovery from inflammatory pain [225]. Attentional deficits are more prevalent in male mice subjected to spared nerve injury (SNI) and correlated to loss of parvalbumin in L5/6 of IL, suggesting a potential role of sex differences in affective pain processing for this subregion [89].

## 6. Pharmacology of mPFC Neurotransmitter Systems 

### 6.1. Glutamate

Glutamate serves as the primary excitatory neurotransmitter and as a key neuromodulator to control synapse and circuit function over a wide range of spatial and temporal scales in mammalian brains. This functional diversity is implemented by two receptor families, the ionotropic glutamate receptors (iGluRs) of the AMPA or NMDA type and the metabotropic glutamate receptors (mGluRs) [226]. Increased activity of the mPFC is mainly associated with pain-alleviating effects in chronic pain models and infusion of local anesthetic results in nociception, suggesting tonic analgesic activity of these brain regions [223]. Reduced glutamate in the mPFC is associated with emotional and cognitive dysregulation in people with chronic pain [227]. Correspondingly, microinfusion of glutamatergic agonists as well as a partial NMDA receptor agonist in the PFC or even introducing NMDA precursors to the drinking water reduces pain-like behavior, improves the performance in cognitive tasks and increases social interaction behavior in neuropathic rats [173,228]. Direct injection of AMPA receptor positive allosteric modulators in the PFC has an antinociceptive effect and increases the analgesic effect of morphine [229].

A mechanistic approach towards these pharmacological observations is however complicated by the complexity of circuits and differential contributions of glutamatergic receptors at different neuron populations resulting in diverging reports on the contribution and regulation of iGluRs and mGluRs in neuropathic pain models. Injection of the iGluR agonist AMPA into the PrL/IL increases allodynia, and AMPA antagonists inhibit allodynia [230]. Nerve injuries increase AMPA type iGluR (GluA1) accumulation in postsynaptic densities of ACC layer L5 pyramidal neurons, and GluA1 phosphorylation by PKA and redistribution from the cytosol to the membrane plays a critical role in LTP induction and ACC sensitization as well as mechanical hypersensitivity [75,158,200,231,232]. Not surprisingly, GluA1 inhibition or disruption of GluA1 phosphorylation in the ACC ameliorates mechanical hypersensitivity resulting from nerve injury [158,200]. PKMzeta, an important factor in neuronal synaptic potentiation, regulates AMPA receptor trafficking in ACC layer L2/3 pyramidal neurons and inhibition of PKMzeta reverses the increased amplitudes of synaptic currents and attenuates mechanical hypersensitivity [233,234]. Since GluA1 is a downstream target of PKMzeta, and PKMzeta inhibition alleviates inflammation-induced mechanical hypersensitivity as well as anxiety-related behaviors, presumably by normalizing GluA1 protein levels, this is suggested to underlie the pain-relieving effects of electroacupuncture [235].

In contrast, the role of NMDA type iGluRs in mPFC is controversial: NMDA or NMDA receptor agonists in the PrL induce antinociception in neuropathic pain models, whereas an NMDA antagonist reduces fear-induced anti-nociception [236,237,238]. Peripheral inflammation increases glutamatergic presynaptic release, miniature excitatory postsynaptic current (mEPSC) frequencies, and neuronal sensitization via iGluRs containing the NMDA type subunit NR2B; and the increased probability of glutamate release is necessary for the development of inflammatory pain [231,239,240,241,242,243,244]. Additionally, NMDA iGluR subunits NR2B and NR2A are closely related to aversion induced by inflammatory pain [62,245,246,247]. Several enzymes and kinases such as AC1/cAMP/PKA, ERK, p38 MAPK, or modulators such as Zif268/Egr1 affect the circuits that are relevant for aversive reactions in the ACC [248,249,250,251]. Interestingly, caveolin-1 is involved in the ERK/CREB signaling pathway and NR2B membrane trafficking [252,253]. NR2B is upregulated via BDNF/mTOR signaling and TrkB or mTOR inhibition can prevent ACC hyperexcitability, as well as aversive behavior and cold hypersensitivity [247,254,255]. NR2B in the rostral ACC is involved in neuropathic pain-induced aversion as a downstream target of BDNF, and antagonizing TrkB/mTOR augments mechanical hypersensitivity [256,257]. Together, NR2B inhibition and/or downregulation alleviate hypersensitivity and affective behavior and relieve mechanical and thermal hypersensitivity in rodent models of neuropathic and bone cancer pain [62,232,239,245,246,247,256,258]. However, this is in conflict with the overall analgesic effect reported from NMDA agonists, which improve pain-like behavior and cognitive performance in neuropathic rats [236,237,238]. Similar inconsistencies are emerging for metabotropic mGLuR1 receptors, which are upregulated in L5 pyramidal ACC neurons, as a result of a nerve injury, and induce neuronal excitation via the PKC signaling pathway. mGluR1 knockdown selectively attenuates thermal, however, not mechanical hypersensitivity or escape/avoidance behavior [134]. Acid-sensing ion channel 1a (ASIC1a) may act as a modulator of cortical LTP downstream of mGluR1 since ASIC1a inhibition abolishes thermal and mechanical hypersensitivity, however, does not affect aversive behavior [259]. In contrast to mGluR1, mGluR5 is associated with depolarization-induced suppression of excitation and exhibits anxiolytic effects [260]. 

Several pharmacological tools targeting glutamatergic signaling induce antinociceptive and/or anxiolytic effects in inflammation, presumably by inhibiting the upregulation of GluR1, NR2A, and NR2B in the ACC, as well as by reducing microglia and astrocyte activation and the induction of proinflammatory cytokines [261,262,263]. However, since these potential drugs are so far not directly applied to the ACC, accounting for other effects is challenging and more specific approaches are necessary to target specific neuron populations and their locations within defined brain regions to provide more mechanistic understanding. As a novel and possibly more selective approach intracranial injection of miR-539 mimics reduces NR2B expression and attenuates mechanical hypersensitivity, indicating that microRNAs may be of potential use to target specific genes in neuronal subpopulations in the brain for therapeutic purposes [232]. 

### 6.2. Gamma-Aminobutyric Acid (GABA)

In the mammalian brain, GABA is synthesized primarily from glutamate, loaded into synaptic vesicles by a vesicular neurotransmitter transporter (VGAT) and released from synaptic terminals mainly of local inhibitory neurons in a calcium-dependent manner. Ionotropic GABA_A_ receptors carry chloride, however, whereas GABA_A_ receptors normally are composed of combinations of several subunit types, GABA_A_-ρ receptors are composed of only ρ-subunits [264]. Metabotropic GABA_B_ receptors are associated with G proteins (for review see [265]). Chronic pain leads to an increased input onto mPFC GABAergic interneurons that can inhibit pyramidal neurons, which in turn shuts down the descending inhibitory efferent pathways [26,88,160]. An example is the input from the thalamic paraventricular nucleus to inhibitory neurons of the mPFC, which is enhanced in a rat model of visceral pain [170]. Correspondingly, inhibition of this input attenuates pain through descending pain modulation. Inflammatory pain induces a BLA driven increase in levels of GABA in the PrL, resulting in reduced excitability of pyramidal neurons, and application of the GABA_A_ receptor blocker bicuculline reverses inflammatory mechanical hypersensitivity [211]. Additionally, neuropathy deactivates PrL pyramidal neurons as a result of a feed-forward inhibition via FS-PV+ GABAergic interneurons, and optogenetic inhibition or activation of inhibitory circuits in the mPFC alleviates pain or increases the nocifensive behavior [88,160].

Neuropathic pain enhances the expression levels of GABA transporter-1 (GAT-1), GABA_A_ subunits β2, β3, δ, and γ2, as well as GABA_B_ subunit β2 in the ACC [266,267], and GABA_A_ and GABA_B_ receptors appear to target different components of neuropathic pain. Inhibition of GABA_B_ in the ACC increases sensitivity to mechanical stimuli in naive animals, whereas activation ameliorates nerve injury-induced mechanical hypersensitivity [268]. Surprisingly, activation of GABA_A_ receptors in PrL and IL increases mechanical but not thermal pain sensitivity [269]. In the rostral ACC, microinjection of GABA_A_ but not GABA_B_ agonists attenuates escape/avoidance behavior induced by nerve injury or peripheral inflammation, however without affecting mechanical hypersensitivity [270,271]. PrL GABA_A_ receptor agonists impair the acquisition of formalin-induced aversive behavior, which suggests a specific role of PrL for affective pain processing [272].

As a consequence of peripheral inflammation, altered synaptic inhibitory transmission develops due to impairment of the GABAergic presynaptic vesicle membrane fusion machinery, which may explain decreased GABA transmission [147]. Finally, apoptosis of inhibitory GABAergic interneurons may occur as a consequence of peripheral nerve lesion and could add to the reduction in inhibitory/excitatory ACC connections as well as the gray matter loss associated with SNI-induced mechanical hypersensitivity [79,145,273]. 

### 6.3. Endogenous Opioids

mPFC disinhibition can also be triggered by the activation of opioid receptors expressed on GABAergic interneurons. Clinical studies show decreased opioid binding in the contralateral PFC of patients with peripheral neuropathic pain [274]. This indicates increased endogenous opioid release, which leads to receptor internalization or even neuronal loss [275]. In a rat model of neuropathic pain, morphine in the ventrolateral PFC alleviates mechanical allodynia in a dose-dependent manner, and this effect is blocked by the opioid receptor antagonist naloxone [276]. The opioid-induced inhibition in the PFC is mediated mainly via µ-opioid receptors, but expression of kappa-opioid receptors and prodynorphin mRNA is increased in the PFC of neuropathic mice and thus kappa-opioid receptors may be relevant likewise [277]. 

### 6.4. Noradrenaline (NA)

The majority of noradrenergic terminals in the cortex originate in the locus coeruleus in the midbrain. NA release from neurons arriving from locus coeruleus at the mPFC suppresses nociceptive-evoked activity in mPFC via alpha2B adrenergic receptors to attenuate neuropathic mechanical hypersensitivity [278,279,280]. Interestingly, neuropathic pain induces noradrenergic impairment and locus coeruleus neurons exhibit increased bursting activity, increased expression of enzymes of noradrenergic metabolism and transport as well as increased expression and sensitivity of α2-adrenoreceptors [281]. HCN channels are regulated by α2-adrenergic receptors, which highlights the role of these channels and the lowering of cAMP in reducing persistent firing and hyperexcitability of layer L2/3 mPFC pyramidal neurons [72]. In contrast, proalgesic effects appear to be mediated by alpha1 receptors [282]. Released noradrenaline induces persistent firing in pyramidal neurons of the PFC via α1-adrenoreceptors that are located on glutamatergic terminals in the PFC and facilitate fast synaptic transmission [283].

### 6.5. Dopamine (DA)

GABAergic neurons of the mPFC are largely innervated by dopaminergic fibers, and the availability of the D2/D3 receptor subtypes in the PFC correlates with the efficacy of placebo analgesia, suggesting a role of the dopaminergic neurotransmitter system in pain processing [284]. Manipulating DA levels in the mPFC can modulate nociceptive responses in a differential, dose-dependent manner. For example, low DA levels in the mPFC are correlated with increased pain sensitivity and exploratory behaviors, while high DA levels reduce pain sensitivity and facilitate stress-related responses, such as escape from threat [285]. Correspondingly, high-frequency stimulation of the dopaminergic input as well as microinjection of a selective dopamine D2 agonist into the ACC/PrL cause long-lasting suppression of nociceptive responses in the PFC [286]. The antinociceptive effect of DA is blocked by the administration of the D2 receptor antagonist sulpiride which suggests D2 specific nociceptive signaling in the PFC, but also highlights the importance of deeper cortical layers, where D2 receptors are mainly expressed [287].

### 6.6. Serotonin (5-HT)

Exposure to a novel environment reduces formalin-induced acute nociceptive behavior, which is accompanied by reduced 5-HT and DA metabolites in the mPFC [288]. Correspondingly, the analgesic serotonin receptor 5-HT1A agonist befiradol reduces the activity of dorsal raphe serotonergic neurons and increases the discharge rates of mPFC pyramidal neurons, which is paralleled by decreased extracellular serotonin and increased dopamine in the mPFC [289]. L5 ACC pyramidal neurons express high levels of HCN channels, which develop dysfunction in neuropathic pain models [72]. Serotonin neurotransmission via 5-HT7 restores HCN function and normalizes mechanical hypersensitivity induced by nerve injury [157,290]. 

### 6.7. Acetylcholine (ACh)

ACh exerts its neuromodulatory function by both synaptic as well as volume transmission [291]. ACh release can be phasic within seconds or tonic lasting for minutes, which would induce short-term or long-term modulations of the mPFC activity depending on the attentional demands [291,292,293]. Two types of ACh receptors (AChRs), the ionotropic nicotinic (nAChRs) and metabotropic muscarinic (mAChRs) receptors, bind ACh. nAChRs are pentameric ligand-gated ion channels that open upon ACh binding, allowing a direct cationic inward current to depolarize the membrane. Depending on subunit composition they show different affinities for ACh binding, a timescale of evoked currents, as well as calcium conductance [294,295,296,297]. mAChRs are G-protein coupled receptors that interact either with G_q/11_ proteins (M1, M3, and M5 mAChRs) or G_i/o_ proteins (M2 and M5 mAChRs) inhibiting adenylate cyclase [298,299].

Cholinergic modulation of the mPFC has been implicated in various aspects of behavior and cognition, particularly arousal, learning, and attention [20,293]. Cortical gamma band oscillations (20-80 Hz) are a representation of cortical synchrony underlying memory formation, and they encode tonic pain perception in the mPFC [300,301]. These oscillations can be induced by cholinergic agonists, suggesting the role of cholinergic signaling in oscillatory activity patterns [302]. While cholinergic signaling under physiological conditions increases the excitability of mPFC neurons, impaired cholinergic signaling appears to contribute to the deactivation of the mPFC and, therefore, appear to be responsible for the deficits found in chronic pain patients. Accordingly, cholinergic modulation of PrL layer L5 pyramidal cells is impaired after nerve injury and this appears to be related to reduced surface expression of M1 mAChRs in PrL layer L5 (Table 1) [155]. 

Chemogenetic inhibition of medial septum cholinergic neurons abolishes the hyperexcitability of rostral ACC pyramidal neurons, reduces CPA, and alleviates inflammation-induced mechanical and thermal hypersensitivity, whereas activation of the same medial septum neurons abolishes hypersensitivity by activating the ventral hippocampus [50]. Interestingly, the M1-induced increase in mechanical thresholds observed in injured rats, is also enhancing GABA_A_ mediated transmission in the ACC [303,304]. Although the importance of ACh and cholinergic signaling in the mPFC has been clinically recognized, since for example the anti-depressive effect of the muscarinic antagonist scopolamine is mediated by PrL/IL M1 receptors, neither the (micro-)circuitry of cholinergic control nor the mechanisms underlying the SNI-induced downregulation of muscarinic receptors at PrL pyramidal neurons are fully understood so far [155,293,305]. 

### 6.8. Endocannabinoids

Mice with induced osteoarthritis exhibit elevated levels of the endocannabinoid 2-arachidonoylglycerol in the PFC [308]. As previously discussed, the excitability of PFC glutamatergic neurons is inversely related to amygdala activity during painful experiences. Coactivation of CB1 cannabinoid receptors and mGluR5 in the IL of arthritic rats inhibits the pain-related increase of amygdala output neuron activity [22]. Correspondingly, systemic application of the endocannabinoid palmitoylethanolamide reverses changes in the glutamatergic synapses in the mPFC, and resolves pain and depressive-like symptoms associated with neuropathic pain [104]. In complete Freund’s adjuvant (CFA) treated mice, this process is suppressed and mGluR5 protein levels are reduced, whereas injection of a positive allosteric modulator of GluR5 into ACC restores depolarization-induced suppression of excitation and exhibits antinociceptive and anxiolytic effects [260]. In particular, endocannabinoids acting on CB1 receptors are critically involved in a long-range brain circuit targeting PAG, which alters pain behaviors by reducing descending noradrenergic and serotoninergic modulation of spinal pain signals [161]. Interestingly, TRPV1 channels, which can be activated by endocannabinoids, modulate pain processing in the brain [309]. Neuropathic pain induces upregulation of TRPV1 together with fatty acid amide hydrolase (FAAH), and single acute microinjection of combined TRPV1/FAAH antagonists into PrL/IL restores aberrant cortical neuron activity [174], whereas microinjection of the TRPV1 agonist capsaicin decreases allodynia [230].

## 7. mPFC Output Regions and Descending Modulation

Common brain regions targeted by projections from both PrL and IL are medial thalamic nuclei, the anterior piriform cortex, olfactory forebrain areas, and the orbitomedial, insular, and entorhinal cortices, as well as the midbrain dopaminergic ventral tegmental area (VTA) and medial regions of the periaqueductal gray (PAG) [310]. In addition, dense reciprocal connections exist between the PrL and IL, as well as the ACC [42,311,312]. However, PrL and IL differ in their specific connections to certain subregions and nuclei within those brain areas [310]. Only PrL connects to the agranular insular cortex, the olfactory tubercle, and the nucleus accumbens, the paraventricular, mediodorsal, and reuniens nuclei of the thalamus, the capsular part of the central nucleus and the basolateral nucleus of the amygdala, as well as the dorsal and median raphe nuclei of the brainstem. The lateral septum, the bed nucleus of the stria terminalis and preoptic nuclei, the substantia innominate and the endopiriform nuclei of the basal forebrain, the medial, basomedial, central, and cortical nuclei of the amygdala, different hypothalamic nuclei, including the supramammillary nuclei, as well as the parabrachial and solitary nuclei of the brainstem are exclusively targeted by IL cortex [310,313]. Augmented synaptic connections mediate a feed-forward inhibition of projections from the PFC to the ventrolateral PAG region and its downstream target regions, and these connections modulate pain behaviors by weakening the descending noradrenergic and serotoninergic inhibitory modulation of spinal pain signals [161]. Regarding projections to the nucleus accumbens, neurons from the PrL project to both the core and shell subregions, whereas IL projections only innervate the medial portion of the shell subregion [310,314]. This difference in output systems might be functionally relevant, constituting a dichotomous “Go/Stop” system for reward-seeking and fear-related behaviors [315]. 

## 8. Summary

In contrast to acute pain, chronic pain serves no biological or protective function but rather represents a disease by itself, and in many cases effective treatments for chronic pain are not available. Chronic pain reduces the quality of life of patients and leads to comorbidities such as anxiety, depression, insomnia, and cognitive disturbance. The relevant affective and cognitive components of the pain sensation are differentially processed in the different subregions of the mPFC by a complex array of inputs targeting the local circuits via multiple neurotransmitter systems (Table 1). Subregion-specific alterations in feed-forward inhibition of pyramidal neurons, and morphological changes of both interneurons and pyramidal neurons arise in different layers of those subregions and contribute to ACC hyperexcitability in contrast to a partial PrL shutdown (Figure 2). The cholinergic system appears particularly relevant since cholinergic projections from the basal forebrain contribute to the cognitive and emotional impairments induced by pain, and inhibition of cholinergic input from the medial septum projecting to the ACC alleviates pain-induced anxiety. While cholinergic signaling under physiological conditions increases excitability in the mPFC, impaired cholinergic signaling appears to contribute to the deactivation of the mPFC and, therefore, may be partially relevant for the deficits found in chronic pain patients. Furthermore, the basis of placebo induced analgesia or expectancy of pain relief is based on the ACC-PFC-PAG descending pain pathways. These predominant roles in affective pain processing put the medial prefrontal cortex in a pole position as a hub for the development of mental comorbidities associated with chronic pain and a relevant target region for therapeutic interventions.

## Figures and Tables

**Figure 1 ijms-21-03440-f001:**
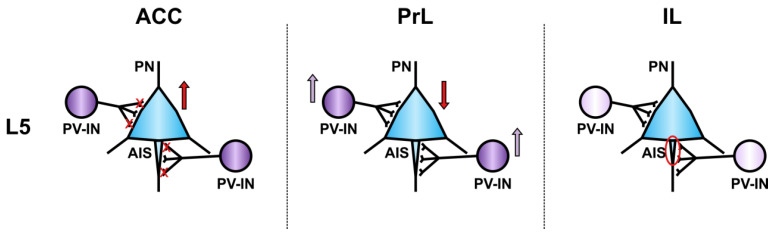
Subregion-specific modification of mPFC L5 pyramidal neurons by PV+ GABAergic interneurons in pathological pain. In the ACC, the number of synapses formed by PV+ interneurons (PV-IN, purple) is decreased (red crosses), which leads to disinhibition of pyramidal neurons (PN, blue) and hyperexcitability; in the PrL, PV+ interneurons increase their feed-forward inhibition onto pyramidal neurons, thereby decreasing pyramidal neuron excitability; in the IL, PV content is decreased (faint purple) in PV+ interneurons and axon initial segment (AIS) of pyramidal neurons is shortened (red circle), which has been associated with decreased pyramidal neuron excitability [90].

**Figure 2 ijms-21-03440-f002:**
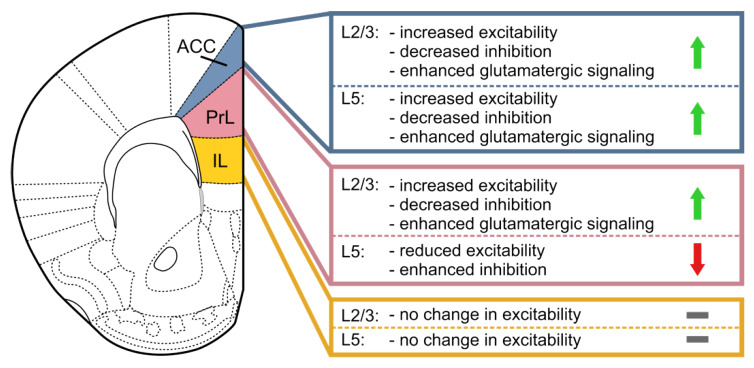
Schematic of pain-induced changes of L2/3 and L5 excitability in the mPFC subregions. Green arrows indicate a net increase in excitability, red arrow indicates a net decrease in excitability, gray minuses indicate no reported excitability changes.

**Table 1 ijms-21-03440-t001:** Regulation of neurotransmitter receptors and ion channels in the different subregions and layers of the mPFC.

Pala	Region (Layer)	Regulation	Pain Model	Reference
AChR M1	PrL/IL (L5)	down	SNI	[155]
AMPA GluA1	ACC	up	CFA-induced chronic inflammatory pain	[243]
AMPA GluA1	ACC	up	CFA-induced chronic inflammatory pain	[261]
AMPA GluA1	ACC	up	CFA-induced chronic inflammatory pain	[262]
AMPA GluA1	ACC	increased membrane insertion	peripheral nerve ligation	[200]
AMPA GluA1	ACC	increased membrane insertion	peripheral nerve ligation	[158]
GABA_A_ β2	ACC	up	Paclitaxel-induced thermal hyperalgesia	[266]
GABA_A_ β3	ACC	up	Paclitaxel-induced thermal hyperalgesia	[266]
GABA_A_ δ	ACC	up	Paclitaxel-induced thermal hyperalgesia	[266]
GABA_B_	ACC	up	Paclitaxel-induced thermal hyperalgesia	[266]
HCN1	ACC	down	CCI	[134]
HCN	ACC/PrL (L5)	functional reduction of I_h_	SNI	[72]
HCN	ACC/PrL/IL (L2/3)	I_h_ modulation by noradrenergic alpha2 activation	SNI	[140]
HCN	ACC (L5)	functional reduction of I_h_	CCI	[157]
mGluR1	ACC	up	CCI	[134]
mGluR5	ACC	down	CFA-induced chronic inflammatory pain	[260]
mGluR5	PrL	up	SNL	[220]
mGluR7	PrL	down	SNI	[219]
Na_v_1.1	ACC	up	Paclitaxel-induced thermal hyperalgesia	[306]
Na_v_1.2	ACC	up	Paclitaxel-induced thermal hyperalgesia	[306]
Na_v_1.6	ACC	up	Paclitaxel-induced thermal hyperalgesia	[306]
Na_x_	ACC	up	Paclitaxel-induced thermal hyperalgesia	[306]
Na_v_ β1	ACC	up	Paclitaxel-induced thermal hyperalgesia	[306]
Na_v_ β3	ACC	up	Paclitaxel-induced thermal hyperalgesia	[306]
NMDA NR2A	ACC	up	formalin-induced conditioned place avoidance	[307]
NMDA NR2A	ACC	up	CFA-induced chronic inflammatory pain	[261]
NMDA NR2A	ACC	down	CFA-induced chronic inflammatory pain	[263]
NMDA NR2B	mPFC	up	SNI	[173]
NMDA NR2B	ACC	up	CFA-induced chronic inflammatory pain	[239]
NMDA NR2B	ACC	up	formalin-induced conditioned place avoidance	[307]
NMDA NR2B	ACC	up	CFA-induced chronic inflammatory pain	[261]
NMDA NR2B	ACC	up	CFA-induced chronic inflammatory pain	[262]
NMDA NR2B	ACC	up	CFA-induced chronic inflammatory pain	[263]
NMDA NR2B	ACC	up	SNI	[247]
TRPV1	PrL/IL (L2/3)	up	SNI	[174]
TRPV1	PrL/IL (L2/3 and L5)	up	SNI	[230]

CCI, chronic constriction injury; CFA, complete Freund’s adjuvant; SNI, spared nerve injury; SNL, spinal nerve ligation.

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
