# Peer review of "The Medial Prefrontal Cortex as a Central Hub for Mental Comorbidities Associated with Chronic Pain"

_ijms, 2020, doi:10.3390/ijms21103440_

Round 1

Reviewer 1 Report

This manuscript by Kummer et al summarizes the structure, types of neuro, circuitry, input pathways, and the specific roles of the medial prefrontal cortex in chronic pain-induced mental comorbidities. Furthermore, they discuss the pharmacology of various neurotransmitters in mPFC. Overall, this is a great review article.

Author Response

We thank Reviewer 1 very much for reviewing our manuscript and for this very positive comment

Reviewer 2 Report

This review paper aims to dissect the main local circuitry, including transmitter systems, together with input and output connectivity of medial prefrontal cortex (mPFC) sub-regions and their importance for affective pain processing and the mental comorbidities frequently associated with chronic pain disorders. Overall the paper is very well written and the message is clear. The findings would definitely be of interest to the readership of the journal. If possible, the authors should consider adding a figure/cartoon that outlines what we know/don’t know about mPFC. That I think would improve the paper and summarize the findings into a single take home visual. Errors in typographical or formatting must be corrected.

Author Response

We thank Reviewer 2 very much for reviewing our manuscript and for their valuable suggestion to add a summarising figure.

Accordingly, we now provide as Figure 2 a graph depicting the main excitability changes induced by pain in the different subregions and layers of the mPFC according to this reviewers suggestion. Also, we checked the manuscript for typographical and formatting errors and corrected the section numbering.

Reviewer 3 Report

In this manuscript entitled "The medial prefrontal cortex as a central hub for mental comorbidities associated with chronic pain" authors reviewed current literature on cortical area involvement on chronic pain and speculate on potential therapeutic intervention on such areas for pain treatment.

Some points that authors should consider are the following:

  • I found the review pleasant to read, I would suggest to reduce the excessive anatomic description in the section "Input systems to mPFC" and "Neuron types and circuitry in mPFC cortical layers";
  • I suggest trying to balance the review (which is very comprehensive) including in the review the crucial role of glial cells in modulating neuronal functions, also speculating on potential targets modulating glial functions;
  • In the section 3.1.6 authors mentioned astrocytes and microglia function in modulating electrical and synaptic activity of ACC neurons. I suggest to extend this concept, in particular referring to the central chronicization processes of pain that are also related to others intercellular mechanisms and on milieu conditioning exerted by glial cells. I suggest (among many others) to use the following papers to expand this concept: PMID: 15667933; PMID: 24919967; PMID: 31030416; PMID: 22638776.

Author Response

We thank this reviewer very much for reviewing our manuscript and for the very positive and constructive comments.

Due to the importance of input/output pathways regulating pain chronification we prefer to leave the anatomical descriptions in, since the provide the reader with a strucutred overview of which regions might modulate mPFC-mediated central pain mechanisms.

We agree with this reviewer regarding the role of glial cells and are grateful for this suggestion. Thus, we added an additional paragraph on astrocytes and microglia in the mPFC subregions and how they might participate in circuit-remodelling of the mPFC in pain.

Round 2

Reviewer 3 Report

I found the revised manuscript appropriate for publication.